# Formation, Losses, Preservation and Recovery of Aroma Compounds in the Winemaking Process

**Bozena Prusova \*, Jakub Humaj** 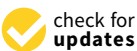**, Jiri Sochor** **and Mojmir Baron**

Department of Viticulture and Enology, Faculty of Horticulturae, Mendel University in Brno, Valticka 337, CZ-691 44 Lednice, Czech Republic; xhumaj@mendelu.cz (J.H.); jiri.sochor@mendelu.cz (J.S.); mojmir.baron@mendelu.cz (M.B.)

\* Correspondence: bozena.prusova@mendelu.cz; Tel.: +420-519-367-259

**Abstract:** A wine's aroma profile is an important part of the criteria affecting wine acceptability by consumers. Its characterisation is complex because volatile molecules usually belong to different classes such as alcohols, esters, aldehydes, acids, terpenes, phenols and lactones with a wide range of polarity, concentrations and undesirable off-aromas. This review focused on mechanisms and conditions of the formation of individual aroma compounds in wine such as esters and higher alcohols by yeast during fermentation. Additionally, aroma losses during fermentation are currently the subject of many studies because they can lead to a reduction in wine quality. Principles of aroma losses, their prevention and recovery techniques are described in this review.

**Keywords:** aroma compounds; fermentation process; aroma losses; esters; higher alcohols

## 1. Introduction

Wine is a complex matrix composed of several hundred chemical compounds or groups such as water, ethanol, glycerol, organic acids, carbohydrates and polyphenols. Volatile aroma compounds, such as terpenoids, pyrazines, higher alcohols, and ethyl esters, play an important role in wines even in small concentrations [1–4].

The primary and secondary metabolites which have been identified in grapes, musts and wines are synthesised throughout several pathways occurring from the vineyard to the consumer [5]. Some of them are produced in grapes; therefore, grapevine variety, vineyard management, geographical region, terroir and climatic conditions are all important factors. Terpenoids and their derivatives constitute significant markers of grape quality, contributing floral notes to the wine flavour and aroma when present in amounts higher than its odour threshold [6,7]. Volatile compounds originating from biosynthesis in grapes give the wine its primary aroma. During the winemaking process, new aromatic compounds are formed as a result of yeast metabolism. This process highly depends on wine microflora that occur in grape must and in conditions of fermentation. During aging, several enzymatic and chemical reactions occur to give wine a tertiary aroma. However, it is not only the complexity of these substances that is responsible for the quality of a wine's aroma. All interactions of odorants with other nonvolatile wine compounds influence the resulting aroma quality, and this is also a field of curiosity in the scientific community. There is a need for a deeper and more comprehensive understanding of the chemistry and biochemistry of fermentation and wine aging [8–12].

The concentrations of volatiles at the end of fermentation depend primarily on their synthesis by yeast but they may also be significantly modified by losses resulting from stripping by $CO_2$ [13]. The loss of aromatic compounds may be significant, affecting the final concentration of volatile aromatic compounds and the sensory profile of wines.

The formation of aroma compounds is an interesting and current topic that has been examined in many studies. They studied the effect of different yeast strains and lactic acid

bacteria on the properties of wine [14–18] and the formation of undesirable substances such as acetic acid or sulphur compounds [15,19–21].

This review provides a comprehensive view of the formation of aroma-active compounds during wine production and describes the loss of aroma compounds during its production that can lead to a reduction in wine quality.

## 2. Formation of Volatile Substances during the Winemaking Process

Volatile compounds derived from grapes include alcohols, esters, acids, terpenes, norisoprenoids, polyfunctional thiols and carbonyl compounds [22]. The production of higher alcohols and esters during fermentation can have a major effect on wine quality. Higher alcohols are undesirable at high concentrations, although in smaller quantities they are thought to positively contribute to the overall quality of wine. Esters are the most abundant aromatic compounds produced by wine yeasts and are the main contributors to the bouquet of young wines. 3-methyl-1-butyl acetate, hexyl acetate and 2-ethyl hexanoate are considered to be the main components of a fruity flavoured wine [23–25].

Primary aromas are grape-derived volatiles originated from biosynthetic pathways occurring in grapes. They often pass through the fermentation process unchanged and are responsible for varietal aromas. The greatest group of volatile molecules is secondary aromas produced through the winemaking process by yeast metabolism during alcoholic fermentation [26].

Higher alcohols are present in higher concentrations but esters have a large number of contributing molecules responsible for the fruity and floral characteristics of wine [27]. Many grape components are known to be depleted and converted to alcohols and esters through fermentation. Thus, the chemical composition of grapes has a high impact on the production of aroma compounds [28]. Yeast can liberate volatile molecules from various grape-derived conjugates such as glycosidically bound and cysteine- and glutathione-conjugated volatile compounds [29,30].

### 2.1. Higher Alcohols

Higher alcohols play an important role in the aromatic profile of a finished wine. They are produced by yeast as a result of amino acid metabolism. After trans-amino reactions, ketonic compounds cannot be released by yeast, so they use an Ehrlich reaction through which corresponding higher alcohols are produced. There are many amino acid precursors (Table 1), but some higher alcohols do not have any amino acid precursors, so it is assumed that they are formed from intermediates of the TCA cycle [31].

**Table 1.** Higher alcohols and their precursors in wine [32].

| Higher Alcohol | Amino Acid Precursor | Aromatic Notes | Content in Wines | Odour Threshold |
|---|---|---|---|---|
| 2-methyl-butan-2-ol | Isoleucine | Nail polish, solvent malt | 30–100 mg/L | 30 mg/L |
| 3-methyl-butan-1-ol | Leucine | Alcohol notes, nail varnish, solvent amilic notes, malt | 80–300 mg/L | 30 mg/L |
| 2-methyl-propan-1-ol | Valine | Solvent, chemical alcoholic, malt notes, wineosity notes | 50–150 mg/L | 40 mg/L |
| Phenylethanol | Phenylalanine | Floral, rose, honey notes, peach notes | 10–100 mg/L | 10–14 mg/L |
| Methionol | Methionine | Crushed potatoes | 0–5 mg/L | 1 mg/L |
| Propan-1-ol | | Alcohol, ripe fruits | 10–50 mg/L | 306 mg/L |
| Butan-1-ol | | Medicinal | 1–10 mg/L | 150 mg/L |

Ehrlich, Neubauer, Fromherz, Guymon, Sentheshanmaganathan, and Genevois and Lafon were the first investigators to study higher alcohol formation. It was found that higher alcohols can be formed through catabolic and anabolic pathways [33]. Ehrlich was the first to propose that the catabolic pathway of higher alcohols is derived from amino acids, while in the anabolic process they are derived from sugars as byproducts of amino

acid synthesis [34–36]. Ehrlich noticed that higher alcohols were derived from essential amino acids in beverages fermented by yeasts of the genus Saccharomyces. He assumed that the yeasts release $NH_4^+$ from molecules of amino acids and that this was incorporated into yeast proteins while the higher alcohols resulting from this metabolic process were secreted by cells to the environment [37–39]. Ehrlich summarized his findings in the alcoholic fermentation of amino acids theory. He found that 2-methylbutanol, 3-methylbutanol and isobutyl alcohols were obtained through decarboxylation and deamination of leucine, isoleucine and valine, respectively [40].

Neubauer and Fromherz found that the synthesis of higher alcohols from amino acids was more complex. According to their study, amino acids added to the fermentation medium were at first converted to corresponding keto acids then subsequently metabolised to alcohols. Their theory was based on the formation of hypothetical amino acids (through oxidation) which was further deaminated to the corresponding keto acid. The latter was subsequently decarboxylated and reduced [37,40].

*2.2. Esters*

Esters in wine are present in both acetate and ethyl ester form. They are mainly produced by yeast metabolism through fatty acid acyl- and acetyl-Coenzyme A (CoA) pathways. CoA is a critical cofactor for a large number of metabolic pathways and is used to activate intermediates during the biosynthesis of medium chain fatty acids (MCFAs). Ethyl esters are produced by the esterification of ethanol and acyl-CoA intermediates as a result of esterase and transferase enzyme activity. The second group, acetate esters, are the result of condensation reactions between acetyl-CoA and higher alcohols produced by yeast from amino acid metabolism [24,41,42]. Both acetate and ethyl esters have different fruity perceptions in wine (Table 2).

**Table 2.** Aromatic notes of individual esters [43,44].

| Esters | Aromatic Note | Detection Threshold |
| --- | --- | --- |
| Ethyl butanoate | Pineapple, strawberries | 20 µg/L |
| Ethyl hexanoate | Green apples, strawberries, blackberries | 14 µg/L |
| Ethyl octanoate | Floral, fruity, soap | 2 µg/L |
| Ethyl decanoate | Floral, fruity, soap | 200–500 µg/L |
| Ethyl acetate | Unpleasant, solvent, fruity | 12–14 mg/L |
| Butyl acetate | Banana, floral, fruity | 1 mg/L |
| Ethyl propanoate | Cherries | 10 µg/L |
| 2-methylbutyl acetate | Fruity | 5 µg/L |
| 3-methylbutyl acetate | Bananas, ripe apples, candy | 2000–3000 µg/L |
| 2-phenylethyl acetate | Rose, fruity | 2000 µg/L |
| Hexyl acetate | Pears, plums, bananas, currants | 15 mg/L |

In general, ester production is affected by the availability of substrates and enzyme activity in yeast. For acetate ester formation, the presence of two substrates, acetyl-CoA and higher alcohol, determines the nature of the acetate ester formed. Availability of the cosubstrate acetyl-CoA may play an important role as the main limiting factor; some studies show that its concentration can be a limiting factor for their production. Levels of acetyl-CoA may be affected by temperature, fatty acid addition, nitrogen source and the presence of oxygen. Oxygen, nitrogen sources and lipids promote yeast growth and thus the usage of acetyl-CoA, leaving less acetyl-CoA available for ester production [45,46].

On the other hand, several studies found that levels of acetyl-CoA were not affected by fermentation conditions. Additionally, this model does not explain the influence of glucose or nitrogen addition and the lowering of top pressure, three factors that raise both yeast growth and ester production [47].

The availability of higher alcohols as a cosubstrate may be the second limiting factor for acetate ester synthesis. Results of some studies showed that supplementations of 3-methyl butanol to both normal and high-gravity worts increased the production of isoamyl acetate, the corresponding acetate ester [48,49]. Overproducing certain higher alcohols also shows a clear increase in the synthesis of the respective acetate ester [50,51]. These results indicate that the availability of higher alcohols influences the production of the corresponding esters.

For ethyl ester production, the presence of specific MCFAs that are esterified with ethanol determines the nature of the ethyl esters formed. MCFA intermediates may be prematurely released from the cytoplasmic fatty acid synthase (FAS) complex where the MCFAs are synthesised. The control mechanisms operating on fatty acid synthesis are also involved in the control of MCFA formation [52,53]. The key enzyme in the regulation of fatty acid biosynthesis is acetyl-CoA carboxylase [54,55]. During fermentation, long-chain saturated fatty acids accumulate and start to inhibit acetyl-CoA carboxylase, which causes the release of unfinished MCFAs from the FAS complex [56,57]. Results show that the overexpression of the FAS1 and FAS2 fatty acid synthetic genes trigger more MCFA formation [58].

A second parameter affecting ester production is the activity of different enzymes [59]. The best characterised enzymes involved in ester production are the alcohol acetyl transferases I and II (AATase I and II; EC 2.3.1.84), which are encoded by the genes ATF1 and ATF2 [59–62]. The enzymes ATF1p and ATF2p are partially responsible for isoamyl acetate and ethyl acetate production [62–64]. Other enzymes involved in ester production are Lg-ATF1p, an AATase found in lager yeast that is homologous to ATF1p; and EHT1p (ethanol hexanoyl transferase), an enzyme able to catalyse the formation of ethyl hexanoate [56,64–66].

For ethyl esters, their production in yeast is catalysed by two acyl-CoA transferases: ethanol O-acyltransferases EEB1 and O-acyltransferase EHT1, where the former is the main enzyme while the latter plays a minor role. Whereas deletion of both EEB1 and EHT1 resulted in severely decreased ethyl ester production, overexpression of the EEB1 or EHT1 gene from laboratory yeast in a laboratory strain did not result in increased production of ethyl esters. On the other hand, overexpression of EHT1 has recently been shown to increase ethyl ester production [56,67–69].

In conclusion, the amount of esters produced by yeast depends on factors that regulate the amount of CoA in wine (Figure 1). There is also evidence for the association of some specific amino acids with CoA biosynthesis [70–72]. The level of gene expression is not the limiting factor for ester production. The availability of MCFA precursors plays an important role along with acyl transferase enzymes in ester formation by yeast [67].

The balance between ester-synthesising enzymes and esterases, which hydrolyse esters, also plays an important role [73]. In finished wine, a decrease in ester concentration during storage may also be caused by residual esterase activity [46].

### 2.3. Releasing Ester from Yeast

Aroma-active esters are formed intracellularly by fermenting yeast cells. Being lipid-soluble, acetate esters rapidly diffuse through the cellular membrane into the fermenting medium. Unlike acetate esters, the proportion of fatty acid ethyl esters transferred to the medium decreases with increasing chain length: 100% for ethyl hexanoate, 54–68% for ethyl octanoate and 8–17% for ethyl decanoate. Longer-chain fatty acid ethyl esters all remain in the cell. It also seems that the distribution of esters between mediums and cells is dependent on the yeast species used, with a higher proportion of the esters formed remaining in the cells of lager yeast (*Saccharomyces carlsbergensis*). Moreover, the distribution of fatty acid ethyl esters between cells and medium is temperature-dependent; more of each ester is retained at lower temperatures [74–76].

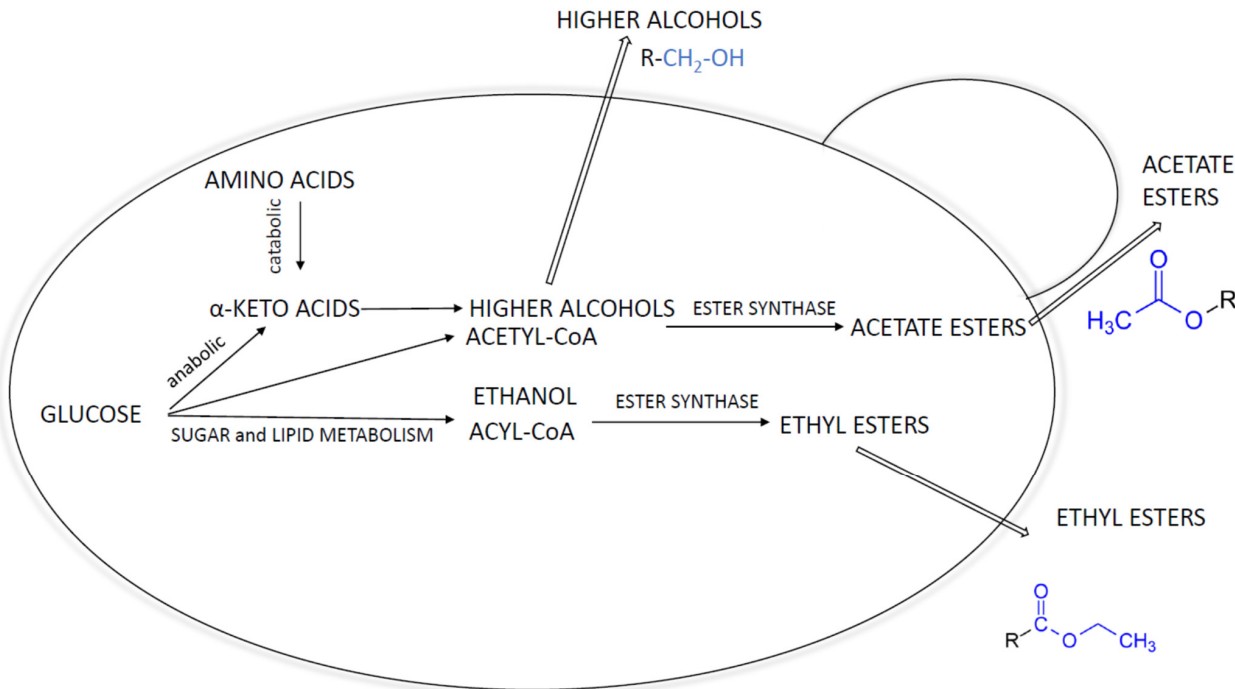

**Figure 1.** Formation of higher alcohols and esters.

### 2.4. Sulphur Compounds

The wine yeast *Saccharomyces cerevisiae* is responsible for the production of several volatile sulphur compounds that have impact on the sensory quality of wine. Important volatile sulphur compounds found in wine are: (1) hydrogen sulphide (rotten egg aroma); (2) methanethiol (methyl mercaptan, cooked cabbage aroma); (3) dimethyl sulphide, dimethyl disulphide and dimethyl trisulphide (cabbage, cauliflower and garlic aromas); (4) methyl thioesters (S-methyl thioacetate, S-methyl thiopropanoate and S-methyl thiobutanoate; cooked cauliflower, cheesy and chives aromas); and (5) the 'fruity volatile thiols' in wine (passionfruit, grapefruit, gooseberry, guava and box hedge aromas) [25,77].

Amongst the volatile compounds produced by yeast metabolism, volatile sulphur compounds represent around 13% of total volatile compounds [78]. Volatile sulphur compounds play an important role in the aromas of foods and beverages, not only because of their broad presence, but also for their significant sensory contributions due to concentrations that are well above their low odour detection thresholds [79,80].

Thiols

Volatile thiols can be found in many foods and beverages including wine, beer, cheese, olive oil, coffee, fruit, meat and vegetables [81]. Potent volatile thiols are highly unstable small molecules that are present at low concentrations with diverse chemical structures. The sulfhydryl (-SH) group in thiols is one of the most reactive functional groups found in natural organic matter [82].

Three varietal thiols are particularly important for wines: 3-mercaptohexyl acetate (3MHA), 3-mercaptohexan-1-ol (3MH) and 4-mercapto-4-methylpentan-2-one (4MMP), which account for passionfruit, grapefruit and box tree (cat's pee) aromas, respectively [83,84]. These varietal thiols are initially not present in grape juice and develop during fermentation by the action of yeast on juice precursors [83,85].

During wine fermentation, the assimilatory reduction of sulphate by wine yeast (to biosynthesise cysteine and methionine) can lead to the excessive production of the $HS^-$ ion, which leads to the formation of $H_2S$ in wine [25,86–88]. This is probably one of the most common problems in a winery, and if not treated, the resulting wine will be tainted leading to a loss in quality and the possibility of rejection by consumers. Wines after fermentation

are often treated with copper sulphate that readily reacts with sulphur compounds to form stable complexes, thereby eliminating the effect of $H_2S$ and mercaptans. However, the use of copper sulphate in wine is not desirable [89–92].

$H_2S$ can be metabolically formed by wine yeast from either inorganic sulphur compounds such as sulphate and sulphite or organic sulphur compounds such as cysteine or glutathionine (Figure 2) [87,89,92,93]. In general, grape must is deficient in organic sulphur compounds, and this can trigger the yeast to synthesise these sulphur compounds from inorganic sources that are usually abundant in grape must. In *S. cerevisiae*, $H_2S$ is the product of the sulphate reduction sequence (SRS) pathway. In the SRS pathway, $H_2S$ is derived from the $HS^-$ ion, a metabolic intermediate in the reduction of sulphate or sulphite needed for the synthesis of organic sulphur compounds. If during fermentation these reactions proceed in the presence of a suitable nitrogen supply, the $HS^-$ ion is sequestered by *O*-acetyl serine and *O*-acetyl homoserine, derived from nitrogen metabolism, to form organic sulphur compounds such as methionine and cysteine [93–95]. However, if nitrogen sources are insufficient or unsuitable, free $H_2S$ can accumulate in the cell and diffuse into the fermenting must [86,90,91,96,97].

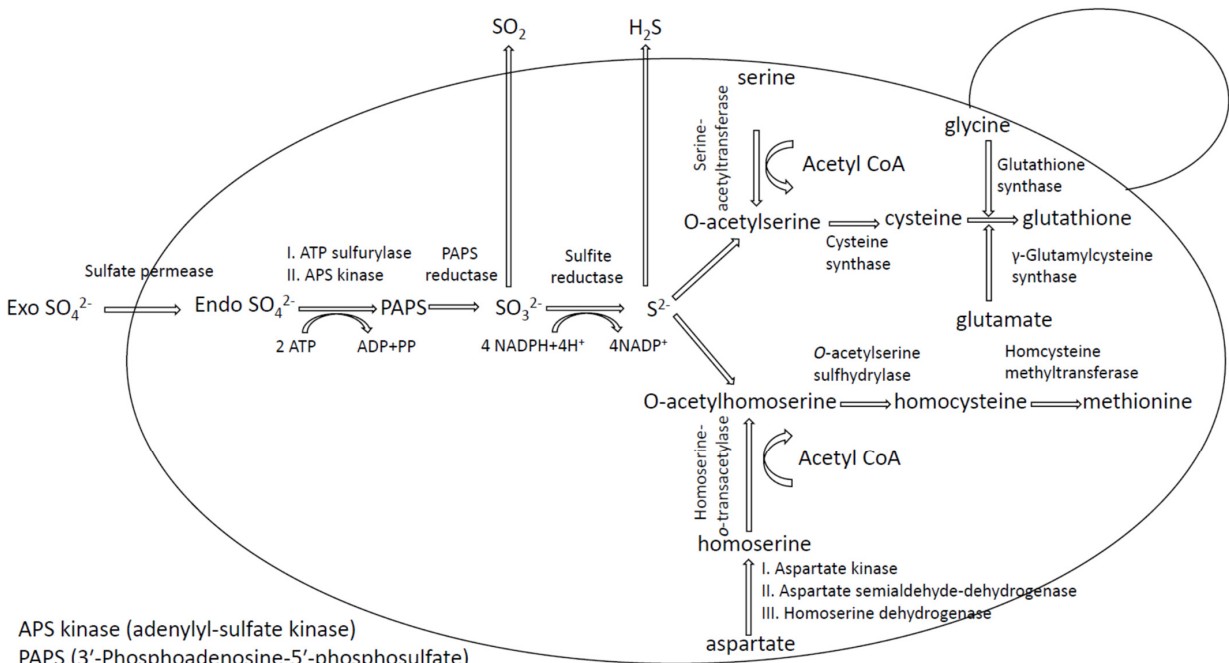

**Figure 2.** Formation of sulphur compound by yeast.

Because the concentrations of cysteine and methionine in grape juices usually are not sufficient to meet the metabolic needs of growing cells, the SRS metabolic pathway is triggered to meet this demand [93]. When adequate nitrogen is present in the medium, sufficient precursors for these amino acids (*O*-acetylserine and *O*-acetylhomoserine) are available to sequester the sulphide. If nitrogen is limited, insufficient precursors are available; the SRS pathway is activated and sulphide accumulates. Sometimes, significant amounts of $H_2S$ are produced when sulphite, which diffuses into the cell, is present in the must during fermentation. Therefore, in conditions of nitrogen depletion, high and continuous production of $H_2S$ is observed in the presence of sulphite [86,92]

It has been proposed that the formation of furfurylthiol from furfural by *S. cerevisiae* is catalysed by the cysteine desulfhydrase enzyme that is required for the production of cysteine. It is through this enzyme that the $HS^-$ anion is produced, resulting in the formation of $H_2S$. The formation of $H_2S$ enhances the formation of furfurylthiol from furfural. This has been confirmed by showing that ferments with an added nitrogen source (thus inhibiting $H_2S$ formation) do not produce as much furfurylthiol. Thus, production

of furfurylthiol is linked to production of the $HS^-$ anion, which is not produced when ammonium sulphate is added to a fermented grape juice in sufficient quantities [98].

### 2.5. Volatile Acidity

Organic acids are significant constituents of juice and wine. Responsible for the sour or acidic taste, they also influence wine stability, colour and pH. The quality and quantity of organic acids in conjunction with sugars has a significant effect on the mouthfeel quality of wines [99]. The wine yeasts are responsible for the production of acetic acid, which also highly impacts the sensory quality of wine. In quantities higher than 0.8–0.90 g/L, it contributes to a vinegar smell and acrid taste, causing the wine to be considered spoiled. Acetic acid is usually formed in small quantities (0.2–0.5 g/L) during alcoholic fermentation as a byproduct of *Saccharomyces cerevisiae* metabolism. Higher concentrations may be associated with contamination by spoilage yeasts and bacteria such as *Candida krusei, Candida stellate, Hansaniaspora uvarum* or *Kloeckera apiculate, Pichia anomala, Saccharomycodes ludwigii, Acetobacter pasteurianus* and *Acetobacter liquefaciens*. Some species of *Oenococcus* and *Lactobacillus* also have the potential to produce acetic acid through the metabolism of residual sugars after malolactic fermentation [31,100].

### 3. Factors Affecting the Composition of Fermentation Gases

A range of factors determine the aroma composition of a wine, including but not limited to grape maturity, grapevine nutrition, harvest method, alcoholic and malolactic fermentation and aging. Among these, grape maturity and alcoholic fermentation are considered the most critical stages [101]. Temperature plays an important role such that cooler temperatures minimize the extraction of phenols without overly affecting the extraction of aroma components [102]. Furthermore, maceration affected pH, total acidity, total nitrogen and amino acid content and potassium, calcium and magnesium levels [103,104]. Some of these parameters may influence fermentation kinetics and contribute to differences in aroma profiles. Compounds related to grape variety and typicity that derive from the berry itself such as monoterpenoids, C13-norisoprenoids, pyrazines and polyfunctional thiols can vary in concentration and can be found either in free form or bound as glycosides or amino acid conjugates, depending on the grape variety [105–107].

The volatile composition of wine is affected by origin and grape variety, which becomes clearer when relative data are used. This fact implies that wine volatile composition data can be effectively used for origin and varietal identification [108]. Variety and origin are two factors with a high degree of interaction, and both have an influence on the volatile composition of the wine, mostly on the volatile compounds related to the amino acid metabolism of yeast and those generated from unsaturated fatty acid peroxidation. Since the volatile composition of wine changes with ageing and strong year-to-year variations are expected due to climatic conditions, the results must be checked with care while taking into account these two additional sources of variability.

### 3.1. Temperature

Fermentation temperature is another variable condition that affects the final concentration of fermentative (secondary) aromas in wine. For example, high concentrations of esters are produced at low temperatures. However, the effect of temperature on the synthesis of higher alcohols is more complex [109]. Molina et al. showed that only the concentration of 2-phenylethanol increases with increasing temperature whereas Beltran et al. observed an increase in the concentration of all higher alcohols [110,111]. Temperature had a partial effect on isoamyl acetate production, so changes in the final liquid concentration of this compound due to temperature can be the result of both evaporation and modification of yeast metabolism. [110–112].

The total ester content in wines is related to the fermentation temperature. In general, total esters were highest in wines produced at low temperatures. Similar behaviour was observed in the content of fatty acid ethyl esters. Several studies have shown that medium-

chain fatty acids (octanoic and decanoic acid) are fermentation inhibitors because they affect cell viability by the reduction of intracellular pH [113,114]. Two mechanisms are capable of reducing the toxic effect of fatty acids, and one of them is esterification. An increase in esterification of fatty acids can reduce the content of carboxylic acids and prevent the occurrence of stuck or sluggish fermentations at low temperatures [42,113]. On the other hand, significant losses of volatile aromas may occur due to volatility and hydrophobicity of each compound, mainly when the temperature of fermentation is high. The presence of fatty acid esters in Merlot wines produced at 25 °C, even in low concentrations, can be related to this phenomenon. Higher alcohols, also called fusel alcohols, have a positive or negative impact on the sensory profile of wines depending on their concentration [115,116].

### 3.2. Nutrition

Several studies have assessed the influence of assimilable nitrogen content on the production of fermentative aromas such as higher alcohols, their acetates and ethyl esters (reviewed in Bell and Henschke 2005). In general, there is a direct relationship between initial nitrogen content and higher (fusel) alcohol concentration when nitrogen content is low, whereas an inverse relationship is found at moderate to high nitrogen quantities [117–121]. Acetates of higher alcohols and ethyl esters show a simpler relationship with nitrogen concentration: an increase in initial nitrogen content is associated with an increase in ester production. Nevertheless, in some cases, the addition of nitrogen can impair the production of esters, depending on yeast strain and the chemical composition of the must [119,122–126]. Overall, Rollero et al. observed that nitrogen concentration had the greatest effect on a large number of volatile compounds. Its simple effect was always positive, i.e., the final concentration of volatile compounds increased along with initial nitrogen content. Temperature and lipid content had moderate effects on some compounds [109].

### 3.3. Yeast

Yeast impacts the aroma composition and therefore the quality of a wine in a number of ways. They synthesise odorant molecules de novo, such as esters and higher alcohols, release odourless precursors and alter wine perception and flavour through the production of ethanol and the release of yeast constituents such as mannoproteins [127]. Different strains of *S. cerevisiae* can produce significantly different flavour profiles when fermenting the same must. This is a consequence of the ability of wine yeast stains to both release varietal volatile compounds from grape precursors and synthesise de novo yeast-derived volatile compounds [77,128–130]. Molina et al. (2009) showed that musts fermented by two strains differed in sensory perception and descriptive analysis [131].

Appropriate selection of the yeast strain not only affects the formation of new aromatic substances such as higher alcohols and esters but can also affect the volatile acid content of wine. *Non-Saccharomyces* yeasts were considered spoilage yeasts in the past. Now, they are used to enhance the aroma profile of wine or to modulate wine composition. Recent publications highlight the role of Non-*Saccharomyces* strains for controlling fermentations mostly in cofermentation with *Saccharomyces*. Some of them showed the ability to not only modulate the total acidity of the wine but also to reduce the volatile acidity values [132].

*Lachancea thermotolerans* can be used to develop a controlled biological deacetification process of wines with high volatile acidity, with the process being oxygen-dependent, which means that its metabolism must shift more towards respiration than fermentation. In refermentation trials, *Lachancea thermotolerans* was able to consume 94.6% of the initial acetic acid [20,133].

*Torulaspora delbrueckii* (formerly *Saccharomyces rosei*) may be used to improve the quality of botrytized wine made using grapes with sugar concentrations up to 350–450 g/L [134]. Higher sugar levels can lead to higher production of volatile acids by *S. cerevisiae* as a stress response [100]. Mixed fermentations with *Zygotorulaspora florentina* and *Starmerella bacillaris* (syn., *Candida zemplinina*) may also have positive effect on volatile acidity in the fermentation of high-sugar musts [135].

*3.4. Skin Contact*

Maceration serves two purposes. Firstly, the juices are extracted and clarified and secondly, maceration favours the dissolution of skin components which improve the quality of wine. Characteristic components of the grape variety (aroma compounds and colloidal substances) are mainly found in grape skin and only reach the must during the prefermentation phase, which indicates that a maceration process takes place [136]. According to Suriano et al. (2015), the maceration time (three, six and eight hours) had a significant effect on the aroma, flavour and colour characteristics of Bombino Nero (*Vitis vinifera* L.) rosé wine [137]. Although a longer maceration time was positively correlated with the colour stability of wine over time, a prolonged maceration was not necessary to enrich the fruity and flowery aroma. The wines macerated for six hours had the highest berry, cherry, exotic fruity, fresh herbaceous and flowery aroma as well as acidity, body and overall judgement scores; therefore, these wines were evaluated as more fragrant and delicate [138].

Macerated wines were also shown to be richer in C6 compounds, particularly 1-hexanol. The effect of C6 alcohols and aldehydes depends on their concentration; at low concentrations (less than 0.5 mg/L), their contribution is positive, adding to the typical aroma of Chardonnay wines, but at high concentrations, they are responsible for herbaceous flavours. The production of fruitier wines can be partly explained by the results of Dennis et al. (2012) which confirmed certain C6 compounds as precursors for hexyl acetate after the respective reduction and/or acetylation [107,139,140].

**4. Fermentation Aroma Losses**

A wine's aroma profile is an important part of the criteria affecting wine acceptability by consumers. Its characterisation is complex because volatile molecules usually belong to different classes such as alcohols, esters, aldehydes, acids, terpenes, phenols and lactones with a wide range of polarity, concentrations and undesirable off-aromas. Furthermore, the nonvolatile wine matrix affects the partitioning of aroma compounds between the matrix and the gas phase depending on their specific chemical properties and interactions with aroma compounds [141–143].

Aroma loss during fermentation is currently the subject of many studies because it can lead to a reduction in wine quality [144]. On the other hand, it has been found that the aroma loss of some volatile compounds can contribute to positive sensory properties in wines. In a consumer study using sensory profiling methods (CATA and projective mapping based on choice), Lezaeta et al. (2017) observed that the perception of white wine quality was improved by increasing positive attributes (good, intense and fruity aroma, apple or pear, etc.) and worsened by removing negative properties (vegetable or herbal and terms always considered negative in the Chilean oenological environment) [145].

During wine fermentation, aromatic complexity increases dramatically [25]. The highest levels of volatile compounds are produced during the yeast growth phase [146,147]. However, some compounds produced during fermentation may negatively affect wine quality, for example, sulphur compounds and other off-aromas [148]. In a study by Muller et al. (1993), different fractions of fermenting gas were assessed [149]. The results of this study showed that only aromatic fractions at the beginning of fermentation were desirable while those produced close to the end of fermentation could negatively affect the aroma of wine. In recent years, there has been a wide range of technologies to reduce aromatic losses, but some of them may negatively affect the kinetics of fermentation, increase production costs due to energy expenditure or increase the risk of stuck fermentations [150]. Fermentation temperature plays an important role in the final aromatic profile [110]. Higher fermentation temperatures contribute to greater losses of volatile compounds while lower temperatures have the opposite effect [149]. Therefore, many winemakers ferment at very low temperatures (8 °C to 12 °C) to reduce aroma loss and improve the aromatic profile of wines [151].

*Aroma Loss Caused by Stripping $CO_2$*

The concentration of volatiles at the end of fermentation depends primarily on their synthesis by yeasts, but it may also be significantly modified by losses into exhausted $CO_2$. Therefore, understanding and modelling the transfer of aroma compounds between the gas and liquid phases would be extremely useful.

Gaseliquid transfer in fermentations has been studied. Unfortunately, the results of these studies cannot be extrapolated to winemaking fermentations. Some studies focused on evaluating the gaseliquid equilibrium in hydroalcoholic solutions and in sugar solutions [152–156]. Models evaluated the transfer of volatile molecules between aqueous solutions and gas phases, which are not applicable to winemaking conditions. The main limitation of these studies is that the concentrations of volatile molecules, as well as the overall composition of the fermenting must, are continuously changing during alcoholic fermentation. A 1996 study by [157] evaluated aroma compound loss due to $CO_2$ release and stripping. Results showed that up to 80% of molecules could be stripped or evaporated from the medium. However, conditions of this experiment were not completely representative of the fermentation conditions of grape must because changes in the matrix composition were not considered and the loss rates were much higher than what is usually observed in the winemaking process. The loss of aromatic compounds may be significant, affecting the final concentration of volatile aromatic compounds and the sensory profile of red wines [158].

## 5. Capture and Aroma Recovery Techniques

The fermentation process is primarily mediated by the yeast *Saccharomyces cerevisiae*, with the basic balance of the process being the conversion of sugars into ethanol and carbon dioxide. A subsequent calculation revealed that 100 g of glucose in the must yields 51.1 g of ethanol and 48.9 g of $CO_2$ [159]. Thus, from 1 L of must at a sugar content of 22.5 °NM and a fermentation temperature of 17 °C, up to 60 L of endogenous $CO_2$ can be produced, which can be efficiently reused through different technological processes. However, thousands of other products are created during fermentation. Some of these remain in the fermentation medium with the substances originating from the must. Many others, however, flow out and are stripped by the flow of the $CO_2$ produced. These substances, together with water vapour and ethanol, form a mixture of fermentation gases. Higher alcohols and aromatics (terpenes and esters) play a particularly important role among the gas-entrained substances. Isobutanol, ethyl acetate, isoamyl acetate and ethyl hexanoate comprise the largest proportion, which give the gas a fruity aroma [160]. Different capture-and-return methodologies have been developed to control aromatic losses and improve wine quality [149,161,162].

Presently, there are many ways to enhance aroma quality. One of the most commonly used methods is the addition of aroma compounds of a different origin. Bioflavours, formed by microbial synthesis, are preferable to chemically synthesised flavours [163]. Chemically synthesised aromas are usually labelled as artificial whereas flavours from microbial fermentation are labelled as nature-identical or natural. Due to their origin, bioflavours often possess greater aromatic diversity than chemically synthesised aromas [164]. In addition to the desired organoleptic characteristics, aromas must be free of harmful chemical contamination resulting from the aroma recovery process.

There are many aroma recovery processes that may have a negative impact on aroma quality. They include distillation, adsorption and solvent extraction, and some of them operate at an elevated temperature or have high-energy consumption. Purification steps to remove solvent residues from final food products also have a negative impact on the environment [165,166].

In recent years, investigations have examined separation processes operating at lower temperatures and without harmful extraction processes, including steam distillation, air stripping the spinning cone column, supercritical carbon dioxide extraction and membrane separation processes [167–171].

The biggest complications that occur during recovery involve the different physico-chemical properties of aroma compounds [164]. In situ recovery from microbial fermentation broth requires a process that is capable of extracting aroma compounds of different chemical natures without disturbing or interfering with fermentation. The recovery process should be efficient at the fermentation temperature, which is usually close to ambient temperature.

One of the promising techniques that can be continuously operated at low temperatures and does not require extraction is pervaporation [172]. The membrane used for pervaporation is nonporous and membrane fouling rarely occurs. Organophilic pervaporation linked to fermentation has been studied separately for the recovery of individual aroma compounds or inhibiting metabolic products and the recovery of individual aromas relevant to the flavour of grapes or wine [173–178].

One of the more complex studies was performed by Schafer et al. in which pervaporation experiments were carried out with samples from a must fermentation to study the possibility of recovering the overall must aroma profile as a concentrate faithful to its origin [171].

According to previous studies, during two independent but organoleptically similar must fermentations, the formation of major compounds in the musts evolved similarly, with only the wine-must concentrations of ethyl acetate differing considerably towards lower must densities. Due to yeast metabolism, the must aroma profile changed during fermentation, which was also reflected in the different qualities of recovered aroma concentrates. An optimum timespan for aroma recovery during fermentation was between densities of 1075 and 1055 g/L corresponding to three to five days. During this time, the complete muscatel aroma profile was recovered as a concentrate and was evaluated as faithful to its origin. At higher ethanol concentrations in the must, esters were greatly enriched in the permeates; whereas the alcohols seemed unaffected, a drastic change in the recovered aroma profile was observed. The effect of ethanol on the fluxes of these esters across the membrane might be due to the formation of hydrogen bonds between ethanol and the esters. This coupling effect can affect the diffusion of esters through the membrane and requires further investigation.

Condensation also seems to be a clean and simple technique, but there are no widespread commercial applications of this technology. This may be due to the fact that most of the technologies have not taken fractioned or characterised condensates during fermentation and from whole aroma condensates with some off-aromas, thereby adding undesirable compounds to wines. In a 2018 study by [145], different aromatic fractions were condensed throughout the fermentation of Sauvignon blanc wines. By applying generic descriptive analysis as described by [179], each condensed fraction was characterised and grouped based on aromatic similarities.

Another method to enrich wine with aromatic substances is supercritical fluid extraction (SFE), which uses fluids at supercritical conditions to selectively extract substances from solid or liquid mixtures. The extraction of aroma compounds by means of supercritical carbon dioxide (SCCO$_2$) can be an attractive alternative to conventional extraction with solvents.

The application of compressed carbon dioxide in the recovery of aroma compounds from fermented beverages is more than 30 years old [180–186]. In wine technology, Jolly was the first to perform an experiment using liquid carbon dioxide to extract aromatic compounds from wines [169,183]; they applied SCCO$_2$ to the extraction of aromas from red wine and coupled the extractor with several separators in series to optimise the aroma fractionation process. In [187], the continuous fractionation of wine with SCCO$_2$ was studied. Several patents were registered on liquid or supercritical carbon dioxide extraction of wines [188,189].

Finally, in a 2008 study by [190], the possibility of extracting aromatic compounds from wine musts by SFE was assessed. Batch high-pressure experiments were carried out where significant enrichment of supercritical extracts on aromatic compounds was observed.

Subsequent high-pressure experiments in a countercurrent SFE column were carried out with muscatel wine and must feeds. The composition of the supercritical extracts confirmed the previous batch experiments. However, there still remains a technical issue that needs to be solved for wider industry use, namely the recovery of aroma compounds from high-pressure stream. The advantages and disadvantages of individual recovery techniques are summarized in Table 3.

**Table 3.** Capture and aroma recovery techniques [145,149,162,171,190].

| Method | Advantages | Disadvantages |
|---|---|---|
| **Pervaporation** | Can be operated continuously at low temperature<br>Does not require any extraction step<br>Does not exert high stress on the active biomass | Membrane fouling of nonporous membrane |
| **Extraction by supercritical carbon dioxide** | Alternative to conventional extraction with solvents | Technical issue with recovery of aroma compounds from the high-pressure stream |
| | Technology for clean chemistry<br>The result is recyclable $CO_2$ and the desired product | |
| **Condensation** | Clean and simple technique | Condensates include some off–aromas—the need of fractionation |
| | | No widespread commercial applications of this technology |
| **Charcoal adsorption traps** | Energy saving | Initial investment and maintenance costs are high<br>Impossible to completely capture all aromatic substances |

## 6. Conclusions

During alcoholic fermentation, many volatile aromatic compounds are formed that can have a positive effect on the quality of the resulting wine. These mainly include higher alcohols and esters with a floral or fruity perception in wine. Their final concentration is affected by many factors such as the availability of individual precursors, yeast strain, grape maturity, grapevine nutrition, harvest method, alcoholic and malolactic fermentation and aging.

The concentration of volatile higher alcohols and esters may be reduced by evaporation or they may be stripped by carbon dioxide during fermentation. This can negatively affect the aromatic appearance of the resulting wine. Techniques to conduct fermentation in order to avoid undesired losses of aromatic substances were described in this review. In addition, various techniques for the recovery of aroma with their advantages and disadvantages were described.

During fermentation of the must, undesirable substances such as sulphur compounds or volatile acids are also formed as byproducts of yeast metabolism. Their amount can be reduced by adding a suitable amount of nutrition and using a suitable strain of yeast. In this regard, the use of non-saccharomyces yeasts, which can reduce the amount of acetic acid when used in coinoculation with saccharomyces yeasts, proves to be effective. Appropriate selection of the yeast strain not only affects the formation of new aromatic substances such as higher alcohols and esters but can also affect the volatile acid content of wine.

**Author Contributions:** Conceptualization—B.P., J.S. and J.H.; writing—original draft preparation, B.P. and J.H.; writing—review and editing, B.P. and M.B.; visualization, J.H. All authors have read and agreed to the published version of the manuscript.

**Funding:** This work was supported by project IGA-ZF/2021-ST2005 Capture of fermentation gas during wine fermentation and project CZ.02.1.01/0.0/0.0/16_017/0002334 Research Infrastructure for Young Scientists.

**Institutional Review Board Statement:** Not applicable.

**Informed Consent Statement:** Not applicable.

**Data Availability Statement:** Not applicable.

**Conflicts of Interest:** The authors declare no conflict of interest.

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
