# Peer review of "Formation, Losses, Preservation and Recovery of Aroma Compounds in the Winemaking Process"

_fermentation, doi:10.3390/fermentation8030093_

Round 1

Reviewer 1 Report

The paper brings together a lot of information already well known in the field and included in many handbooks and university printed courses of oenology. It also adds some newer research results.

In the present form it requires a major revision, especially because of the conclusion Chapter which is not a true conclusion of the paper. It also needs several clarifications and corrections, as well as describing in the text the figures and tables, plus removing the redundant figure 3. It is also advisable to include some references from this journal to demonstrate why a new review is necessary on this topic, when there are already several papers in the Fermentation journal dealing with wine esters, thiols etc.

As the title promises, the paper looks into mechanisms of formation and losses of aroma, but it also extends on techniques for capturing and recovery of aroma. Maybe, but not mandatory, the title can be changed a little to include these aspects too. (Formation, Losses, Preservation and Recovery of Aroma Compounds in Wine?).

The paper is structured enough, but sometimes the discussions get to be too intricate or even unconvincing. Some examples are included hereafter:

  • Lines 263-265 contain a not convincing statement. The fact that the temperature has a partial effect on the accumulation of isoamyl-acetate does not necessarily mean that temperature affects the regulation of its metabolic pathway, unless other proofs can also be presented.
  • Lines 278-279 are confusing. Taking into account the previous phrase, this sentence is probably meaning that the presence of fatty acid esters in Merlot wines produced at 25C, even in low concentrations, could be related to this phenomenon. Otherwise, the way it is formulated, it can be understood that lower than usual fatty acid esters are produced at low temperature, which is not the case.
  • Lines 285-286 saying “higher alcohol concentration” is not clear enough, as it can be interpreted as higher concentrations of alcohol (ethanol), while here the meaning is "the concentration of higher alcohols"
  • Lines 295-296 are not sufficiently elaborated: “some compounds”, “positive or negative effects”… such as which?
  • Starting with line 308 a lot of space is dedicated to a specific experiment with two particular wine yeasts (VIN13 and EC1118). Being a review paper, the discussion in it should be general and broad, thus, either more similar experiments with other yeasts are presented, or the discussion based on these two commercial wine yeasts is removed.
  • Chapter 5. Capture and Aroma Recovery Techniques is difficult to follow and could use some better explanations for each technique. The lines 412-419 are not relevant for wine and should be removed, as addition of aroma compounds of different origins is not allowed in wine, in accordance to the present legislation.

Figures and Tables are simply introduced near the text, but not cited or explained anywhere in the text.

  • In figure 2 some abbreviations are not explained, such as APS kinase (adenylyl-sulfate kinase), PAPS and PAPS reductase (3′-Phosphoadenosine-5′-phosphosulfate reductase), while the rest of the enzymes are described with their full names.
  • While Figures 1 and 2 show some mechanisms of aroma formation, Figure 3 entitled “Aroma compounds in wine” is too general to accurately describe what it represents and why is needed. In the reviewer’s opinion this figure should be removed, as it only shows the sensory effect of some aroma compounds, fact that is already present in Table 1 and Table 2.
  • The entire Table 3 does not have a homogenous style and is not easy to follow. Many statements should be rephrased and shortened. Moreover, the statement (“Usage includes the decaffeination of coffee and tea, hops extraction…”) does not describe an advantage as the column title implies.

The conclusion chapter only focuses on the gas extraction technology for improving wine aroma, thus not being a true conclusion for the entire paper, which is supposed to emphasis the formation and losses of aroma compounds. Moreover, the statement in lines 495-496 is not convincing, as the paper has not really proven that capturing fermentation gas will be more beneficial for the wine and will give positive organoleptic characteristics, but is rather the expectation of the authors or an intention to experiment later on this. The economic effect, included in lines 504-510 is also not a conclusion of this study, but merely a part of the discussion for Chapter 5.

Bibliography is not homogeneously presented, having for example titles entirely with capital letters, with capital letter for each word or as normal sentences.

The Journal Fermentation already has some reviews and original papers dealing with the topic of esters and thiols in the aromatic profile of wines. To demonstrate the difference this paper brings to the journal, other papers from this journal should also be cited.

English should also be carefully revised. The text looks like different sections were written by different people and not proofread in the end.

Several statements or terms are unclear and should be rephrased. To only give some examples:

  • Table 1 – “Crushed otatoes” should be corrected into “Crushed potatoes”
  • Line 127-129 contains a phrase on MCFA which should definitely be rewritten
  • In line 140 in “by two acyl-CoA”, transferase is missing
  • In line 185 “extremely low abundances”, in line 197 “Completed wine ferments”, in line 221 “present in the ferment”, in line 230 “added to a ferment” are not clear
  • In lines 303 and 304 please consider using “different” instead of “difeferential” and “differential”, respectively.
  • In lines 442-448 several terms containing “wine-must” (“wine-must fermentation”, “wine-must concentrations”, “wine-must densities”, wine-must aroma”) are probably referring to “grape must” or simply “must”?
  • In lines 500-501 the statement “the carbon dioxide produced by the yeast is more sensory delicate in taste than the one that is currently produced”, in this form, does not sound right. If it is a purified product to be used in food industry, CO2 is CO2, it cannot be "more sensory delicate" than another CO2.

Author Response

Thank you for reviewing the article and for the very valuable comments that have helped us improve this article. We accepted all your comments and revised the article accordingly.

We have completely rewritten the conclusion. We added references to figures and tables to the text and deleted figure number 3. We have also modified the information in Table 3. Also the quality of figures was improoved.

Although many studies and reviews have been published on this topic, this review provides information not only on the formation of aromatic substances, but also on the possibilities of their recovery. We have added this information at the end of the Introduction section, which also highlights the novelty of this study. Of course, we have also added references to articles from this journal, which dealt with a similar topic.

We have changed the title according to your proposal to: Formation, Losses, Preservation and Recovery of Aroma Compounds in Wine.

We have incorporated all the other comments into the text, as you can see in the attached manuscript.

Reviewer 2 Report

The manuscript deals with the formation and losses of aroma compounds in the winemaking process. The topic is interesting and relevant, however minor changes are needed to make this manuscript acceptable for publication in Fermentation.

General Comments:

According to the authors, the aim of the work was to present a review focused on mechanism and conditions of formation of individual aroma compounds of wine, such an esters and higher alcohols by yeast during fermentation. Fermentation covers high quality original research papers, review articles, short communications and technical notes on all research areas of fermentation related to new and emerging products, processes and technologies. It is a well developed and described study, however there are some important issues that need to be addressed:

1) The authors should highlight the manuscript novelty (clarify it), since there are recent studies covering the same subjective, as follow.

- Management of Wine Aroma Compounds: Principal Basis and Future Perspectives, DOI: 10.5772/intechopen.92973;

- Formation of Aromatic and Flavor Compounds in Wine: A Perspective of Positive and Negative Contributions of Non-Saccharomyces Yeasts, DOI: 10.5772/intechopen.92562.

2) The authors should carefully review spelling and grammatical errors in entire manuscript. For instance, “such a esters” in Abstract section.

3) In Introduction section, when the authors mentioned that “The primary and secondary metabolites, which have been identified in grapes, musts, and wines, are synthesised throughout several pathways occurring from the vineyard to the consumer. Some of them are produced in grapes, so grapevine variety, vineyard management, geographical region, terroir and climatic conditions are all important factors”, they should insert robust references to support the affirmation, such as https://doi.org/10.1016/j.foodchem.2019.03.103.

4) The authors should improve the quality of the Figures 1 and 2, since some words are too small and very difficult to read.

Author Response

Thank you for reviewing the article and for the very valuable comments that have helped us improve this article. We accepted all your comments and revised the article accordingly.

Q1:The authors should highlight the manuscript novelty (clarify it), since there are recent studies covering the same subjective, as follow.

A1:Although many studies and reviews have been published on this topic, this review provides information not only on the formation of aromatic substances, but also on the possibilities of their recovery. We have added this information at the end of the Introduction section, which also highlights the novelty and need of this study. Of course, we have also added references to articles from this journal, which dealt with a similar topic.

Q2:

The authors should carefully review spelling and grammatical errors in entire manuscript. For instance, “such a esters” in Abstract section.

A2: Thank you, it was corrected

Q3:

In Introduction section, when the authors mentioned that “The primary and secondary metabolites, which have been identified in grapes, musts, and wines, are synthesised throughout several pathways occurring from the vineyard to the consumer. Some of them are produced in grapes, so grapevine variety, vineyard management, geographical region, terroir and climatic conditions are all important factors”, they should insert robust references to support the affirmation, such as https://doi.org/10.1016/j.foodchem.2019.03.103.

A3: Reference was added.

Q4: The authors should improve the quality of the Figures 1 and 2, since some words are too small and very difficult to read.

A4: Quality of figures was improved and figure number 3 was deleted.

Round 2

Reviewer 2 Report

The authors properly agreed with the reviewer's observations, as well as correctly changed all the mistakes improving the article quality. Therefore, now I recommend this article for publication in Fermentation journal.